# Nutritional and Behavioral Countermeasures as Medication Approaches to Relieve Motion Sickness: A Comprehensive Review

**DOI:** 10.3390/nu15061320

**Published:** 2023-03-07

**Authors:** Ghazal Rahimzadeh, Abdullatif Tay, Nikolaj Travica, Kathleen Lacy, Shady Mohamed, Darius Nahavandi, Paweł Pławiak, Mohammadreza Chalak Qazani, Houshyar Asadi

**Affiliations:** 1Institute for Intelligent Systems Research and Innovation (IISRI), Deakin University, Geelong, VIC 3216, Australia; 2PepsiCo Inc., Food Safety and Global Process Authority, 433 W Van Buren St., Chicago, IL 60607, USA; 3Food & Mood Centre, IMPACT—The Institute for Mental and Physical Health and Clinical Translation, School of Medicine, Barwon Health, Deakin University, Geelong, VIC 3220, Australia; 4Institute for Physical Activity and Nutrition, School of Exercise and Nutrition Sciences, Faculty of Health, Deakin University, Geelong, VIC 3220, Australia; 5Department of Computer Science, Faculty of Computer Science and Telecommunications, Cracow University of Technology, Warszawska 24, 31-155 Krakow, Poland; 6Institute of Theoretical and Applied Informatics, Polish Academy of Sciences, Bałtycka 5, 44-100 Gliwice, Poland

**Keywords:** motion sickness, treatments, nutrition artificial intelligence

## Abstract

The mismatch in signals perceived by the vestibular and visual systems to the brain, also referred to as motion sickness syndrome, has been diagnosed as a challenging condition with no clear mechanism. Motion sickness causes undesirable symptoms during travel and in virtual environments that affect people negatively. Treatments are directed toward reducing conflicting sensory inputs, accelerating the process of adaptation, and controlling nausea and vomiting. The long-term use of current medications is often hindered by their various side effects. Hence, this review aims to identify non-pharmacological strategies that can be employed to reduce or prevent motion sickness in both real and virtual environments. Research suggests that activation of the parasympathetic nervous system using pleasant music and diaphragmatic breathing can help alleviate symptoms of motion sickness. Certain micronutrients such as hesperidin, menthol, vitamin C, and gingerol were shown to have a positive impact on alleviating motion sickness. However, the effects of macronutrients are more complex and can be influenced by factors such as the food matrix and composition. Herbal dietary formulations such as Tianxian and Tamzin were shown to be as effective as medications. Therefore, nutritional interventions along with behavioral countermeasures could be considered as inexpensive and simple approaches to mitigate motion sickness. Finally, we discussed possible mechanisms underlying these interventions, the most significant limitations, research gaps, and future research directions for motion sickness.

## 1. Introduction

Motion sickness (MS) is characterized by unpleasant symptoms that occur during transportation in real environments, such as by car, train, ship, or plane, or when using virtual environments, such as simulators, movie theatres, video games, or virtual reality (VR) applications [1]. In real environments, MS is caused by the physical movement of the vehicle, where the vestibular and proprioceptive systems provide information about the movement that does not align with the person’s visual perception. On the other hand, in virtual environments, the visual stimulation of motion creates an illusion of movement. However, the vestibular and proprioceptive systems, which provide information about body position and movement, do not experience a corresponding change [2,3]. MS has been classified into two distinct categories. The first type is called transportation sickness, which is also known as real MS and occurs in real environments. The second type is referred to as visually induced motion sickness (VIMS), which occurs in virtual environments [2] (See Figure 1). While these two types of MS are induced differently; however, they both involve a mismatch between the expected sensory and received sensory signals by the brain, which can lead to similar symptoms such as dizziness, headache, blurred vision, salivation, pallor, cold sweating, and nausea and vomiting (See Figure 2). There may be variations in the physiological responses between real MS and VIMS, which can be affected by several factors, including the duration and type of exposure, individual differences, environmental conditions, and dietary intake before the exposure [4,5,6]. Researchers in [1] showed that the symptoms of cybersickness, a subclass of VIMS, tended to be more pronounced than those of real MS.

The mechanisms responsible for producing these symptoms are not well understood and may differ between the two types of motion sickness [7]. The occurrence of MS can lead to alterations in physiological signals of the body, such as an increase or a decrease in heart rate [7], heart rate variability [8], and skin conductance, which measures the electrical conductivity of the skin [9], brain activity [10], and gastrointestinal motility, which is the movement of food through the digestive system [11].

A survey with a total number of 4479 participants showed that 59% of participants had experienced MS at some point in their life [12]. In addition, almost all ship travelers have experienced seasickness under rough conditions [13]. It was initially established that individuals without organs responsible for balance in the inner ears were immune to MS during a sea voyage [14]. Similarly, in [15], Paillard and his colleagues conducted a study on 167 subjects (84 female and 83 male; mean age: 52.9 ± 19.2). Their findings indicated that patients with vestibular impairment were less susceptible to MS compared to healthy subjects, leading them to suggest that the vestibular system plays an important role in the experience of MS [15].

The common theory explaining the cause of MS is known as sensory conflict theory, which proposes that MS can occur due to incompatible visual, auditory, and vestibular sensory inputs. This creates a mismatch between an individual’s perceived and expected internal representation of their spatial surroundings, leading to the symptoms of MS [16]. Therefore, the mismatched signals perceived by the vestibular and visual systems to the brain result in unpleasant MS symptoms [17]. This sensory disparity activates the vestibule-autonomic pathways [17], which have been found to be involved in generating unpleasant MS symptoms such as headaches, cold sweating, pallor, salivation, dizziness, nausea, and vomiting [18,19,20]. In real environments, sensory inputs from body position and movement are combined with visual information to form a coherent perception of the surroundings [21]. This integration enables the brain to comprehend and react to movements and changes in the environment. However, in VR environments, a disconnect between perceived visual signals and the vestibular–proprioceptive signals lead to conflicting sensory inputs, resulting in VIMS [22]. VIMS is believed to be caused by a discrepancy between visual signals and the absence of matching physical movements [2].

Studies have found that the level of sensory conflict between various inputs is positively correlated with the severity of MS [23]. This indicates that greater sensory mismatches are associated with stronger symptoms [24,25]. Consequently, the sensory conflict theory of MS is still widely regarded as the prevailing explanation for the underlying cause of this condition [22].

Numerous neurotransmitters are involved in the activity of vestibular nuclei and may play a role in MS development, including histamine, serotonin, acetylcholine, and dopamine [26,27]

Since the vestibular system is responsible for provoking MS, peripheral or central vestibular blocking agents might be effective in inhibiting MS [28]. However, existing medications can cause numerous side effects [29], such as headache, dry mouth and fatigue [30], dizziness and blurred vision [31], nausea, and vomiting [32], as indicated in Table 1.

Therefore, it is critical to identify alternative approaches, such as behavioral countermeasures or nutritional interventions, to prevent or reduce MS [4,6]. Behavioral and nutritional countermeasures are effective in MS mitigation, as they can be applied in both real and virtual environments [4,62]. The effects of nutrients and meal composition on physiological parameters, such as heart rate signals [63], respiratory rate signals [64], neural signals [65], and myoelectrical signals [66], have been identified, and these physiological signals have been found to be associated with MS [67,68]. So, it is crucial to evaluate both behavioral countermeasures and nutritional approaches in MS studies before considering pharmacological interventions.

The main objective of this paper is to provide a comprehensive review of the effectiveness of pharmaceutical, behavioral, and nutritional interventions for preventing motion sickness (MS), as well as suggestions for future trials. Furthermore, this review aims to describe the underlying mechanisms of these interventions. To the best of our knowledge, this is the first review to investigate the potential benefits of nutrition in combination with all available behavioral and pharmacological treatments for mitigating the symptoms of MS.

## 2. Interventions for Mitigating Motion Sickness

### 2.1. Pharmacological Approaches for Mitigating Motion Sickness

The most common approach to mitigating MS is through pharmacological interventions [50]. However, as shown in Table 1, all existing medications have numerous contraindications and side effects [29,32]. To effectively prevent or decrease the frequency and severity of MS symptoms, it is critical to identify safe strategies [19]. Current MS medications can be categorized into four major groups: (1) Antihistamines [26], (2) Antimuscarinics [69], Sympathomimetics [33], and (4) Corticosteroids [70].

#### 2.1.1. Antihistamines

The central histamine system is involved in several central nervous system functions, such as sympathetic nervous system activation, an increase in stress-related hormones, and MS [27,71]. Therefore, activation of the vestibular system triggers the histaminergic neuronal system during MS occurrence, which eventually stimulates histamine H_1_ receptors in the brainstem and induces vomiting [72]. First-generation antihistamine medications exert their effects by either blocking histaminergic responses mediated by the H_1_ receptors in the nervous system or by inhibiting the activity of histidine decarboxylase (HDC) enzyme, which is the primary enzyme responsible for catalyzing histamine synthesis [73] (see Figure 3).

H_1_-receptor antagonists are frequently prescribed to alleviate symptoms of MS; however, they can lead to unfavorable side effects [31,32]. The most common antihistamine medications for MS treatment are described in the following.

The accidental discovery of dimenhydrinate’s effectiveness in preventing MS occurred when a pregnant woman with severe car sickness tried the medication. She noticed it relieved her symptoms of nausea and vomiting [75]. This finding prompted further investigation and the development of dimenhydrinate as a treatment for MS [76]. Researchers in [77] tested three treatments for MS reduction: (A) three tablets with 20 mg dimenhydrinate each + one placebo tablet; (B) three placebo tablets + one tablet with 50 mg dimenhydrinate; (C) three placebo tablets + one placebo tablet. They found that both formulations of dimenhydrinate (treatments A and B) were effective in preventing MS and reducing the severity of associated symptoms. Additionally, the study found that the level of sedation induced by these two dimenhydrinate formulations was comparable [77]. It has been shown that the mechanism of action of dimenhydrinate is mainly based on inhibiting the vestibular system, which is responsible for our sense of balance and spatial orientation [78]. Dimenhydrinate acts on the vestibular nuclei in the brainstem by diminishing the sensitivity of the vestibular system to motion. This action helps alleviate MS symptoms, including nausea, dizziness, and vomiting [79].

The study conducted by Weinstein and Stern provides valuable insights into the comparative effectiveness of two medications, dimenhydrinate (50 mg) and cyclizine (50 mg), in mitigating MS and its associated symptoms in a group of 23 participants who were exposed to a rotating optokinetic drum [47]. Their study revealed that the protective effect of dimenhydrinate against MS was mainly due to its sedative properties, which helped to reduce the sensation of nausea and vomiting. In other words, the drug’s sedative effects may help to alleviate the symptoms of MS, such as nausea and vomiting, by calming the body and reducing feelings of motion-induced discomfort [29,80]. Cyclizine, on the other hand, was found to be more effective in reducing gastrointestinal symptoms such as stomach discomfort and preventing abnormal contractions of the stomach muscles (known as gastric dysrhythmias) [47].

Promethazine is one of the most extensively administered antihistamine medications for MS prevention [50]. It is commonly used to treat nausea and vomiting, including those caused by MS [81]. The anti-MS effect of promethazine is achieved by competing with histamine for attachment to histamine H_1_ receptors rather than by inhibiting the expression of histamine H_1_ receptors [81]. The use of promethazine as a treatment for space-related MS via intramuscular injections is a common practice by the National Aeronautics and Space Administration Agency, typically administered in doses of 25 or 50 mg [82]. However, a study by Cowings et al. [44] examined the effects of promethazine injections on performance, mood states, and MS. The researchers found that both dosages of promethazine resulted in a significant decrease in performance compared to the placebo. This suggests that the effective doses of promethazine used to treat MS in astronauts could impair their operational performance [44]. It should be noted that the use of promethazine in the treatment of space MS is carefully monitored and controlled. This is due to the medication’s potential to induce sedation and impede cognitive function [83]. Researchers administered promethazine in combination with caffeine to reduce its adverse effects in soldiers transported frequently by helicopter. Their study involved four groups, including promethazine (25 mg) + caffeine (200 mg), meclizine (25 mg), scopolamine (1.5 mg), and an acustimulation wristband. Only the promethazine + caffeine group showed significantly lower side effects, nausea, and MS scores compared to other groups [84]. Promethazine is typically administered by injection, either intravenously or intramuscularly. However, due to the nature of its administration, it may not be suitable for all populations and groups of people [85,86].

Betahistine is a medication that has been shown to interact with multiple histamine receptors in the body, including H_1_, H_2_, and H_3_ receptors [87]. This interaction enables betahistine to have various effects on the body, such as reducing inflammation, improving blood flow to the inner ear, and alleviating symptoms of vertigo [37]. One of the key factors contributing to betahistine’s success in reducing vertigo is its potential to decrease elevated histamine levels in the body [88]. The effectiveness of betahistine in preventing seasickness remains uncertain, as a study found that consuming 48 mg of betahistine did not significantly prevent seasickness [28]. On the other hand, research has shown that 32 mg of betahistine can have a positive effect in reducing MS compared to a placebo [38]. Therefore, further research is still needed to determine the efficiency of betahistine and the exact mechanism of action in preventing MS and other related conditions.

Cinnarizine was first generated as a medication that blocks the H_1_ histamine receptor. However, various studies have shown that the calcium channel blocker activity of cinnarizine is also involved in treating MS and vertigo [89,90,91,92]. In other words, cinnarizine is a medication that works by blocking the movement of calcium ions across the sensory cells in the inner ear, which are responsible for detecting motion and maintaining balance [93]. By inhibiting this movement, cinnarizine can reduce the signals that trigger dizziness, vertigo, and nausea [29,94]. It also helps to maintain the flow of a fluid called endolymph, which is responsible for transmitting signals between the sensory cells and the brain [89]. This is achieved by preventing constriction of the stria vascularis, which is a structure in the vestibular system that helps to regulate the composition and flow of endolymph [89]. By maintaining endolymph flow, cinnarizine can help to reduce symptoms of MS [95,96]. While researchers in [97] reported that cinnarizine had no notable side effects, other investigators in [41] reported drowsiness, dry mouth, and sedation after the administration of cinnarizine. The differences between studies about the efficiency of medications in treating MS can be related to various factors such as dosage of use and different MS stimulus [98,99].

Meclizine is a type of antihistamine that blocks the action of histamine by binding to H_1_ histamine receptors in the brain [100]. This medication is commonly used to treat nausea and vertigo caused by inner ear disorders and MS [100]. Meclizine revealed slower anti-MS effects in comparison to other medications but showed a longer period of action [50]. It has been proposed that this prolonged period of action could be beneficial for the inhibition of MS during long journeys [101]. However, it can lead to side effects such as drowsiness, dry mouth, and constipation [51].

Moreover, it has been suggested that histamine in histaminergic neurons can be reduced by the effects of histidine decarboxylase inhibitors such as α-fluoromethylhistidine (α-FMH) [102]. Histidine decarboxylase is the enzyme that is directly involved in histamine production (see Figure 3) [103]. Thus, α-FMH can block neural transduction of the histaminergic neuron system and diminish the neural histamine content of the brain [26,104]. According to research conducted by Takehiko et al. [104], the histamine level in the stomach and brain declined by taking a single dose of α-FMH, but continual intake of α-FMH reduced the histamine content in all tissues. Overall, antihistamines are considered an effective anti-MS medicine; however, common side effects such as fatigue, drowsiness, dizziness, and impairment of cognitive function, memory, and psychomotor performance have been indicated to be limiting factors of their use [104].

It should be noted that even though the medications mentioned above are categorized as antihistamines, there may be variations in their chemical structures that can impact their interactions with other molecules in the body and the specific symptoms they target [105].

#### 2.1.2. Antimuscarinics

Antimuscarinic agents block the action of muscarinic acetylcholine receptors, which are involved in the regulation of the vestibular system and have been identified as key factors in the development of MS [69,106]. Scopolamine, an antimuscarinic agent, is well-known for its ability to inhibit MS by blocking muscarinic receptors in the vestibular system and other regions of the brain [106,107,108]. The roles of different subtypes of muscarinic acetylcholine receptors in MS are not fully understood and remain an active area of research [107]. Researchers examined the effects of selective muscarinic receptor antagonists on MS symptoms. For example, a previous study by Golding and Stott in 1997 compared the effects of two selective muscarinic receptor antagonists, zamifenacin (20 mg) and scopolamine (0.6 mg), on 18 healthy men who were exposed to cross-coupled stimulation on a turntable. Both compounds were shown to reduce the subjective symptoms of MS, leading the researchers to suggest that compounds with selective M3 and/or M5 antagonism may be effective against MS [109].

A more recent study conducted by Golding and his colleagues in 2018 investigated the effects of two selective M3 receptor antagonists, darifenacin (10 mg or 20 mg) and scopolamine (0.6 mg), on 16 male volunteers who were exposed to cross-coupled stimulation on a turntable. The results of this study revealed that scopolamine was more effective than the placebo in reducing MS symptoms, while the effect of darifenacin at either dose was not statistically significant. As a result, the researchers concluded that selective M3 receptor antagonism might not be necessary for preventing MS [107].

Despite the effectiveness of scopolamine in mitigating MS, it can lead to negative side effects such as memory and attention issues and, in severe cases [110], hallucinations [31]. As a result, further research is needed to better understand the mechanism of scopolamine in preventing MS and to develop safer alternatives with fewer side effects.

#### 2.1.3. Sympathomimetics

Sympathomimetic drugs, such as dextroamphetamine, are not typically used as a first-line treatment for MS [111]. However, they can enhance the effects of other drugs and mitigate their adverse reactions [112]. The potential mechanism behind dextroamphetamine in reducing the side effects of anti-MS medication is not fully understood [113,114]. Researchers in [50] demonstrated that combining scopolamine (at doses of 1 mg and 1.2 mg) with 10 mg of dextroamphetamine is more effective in preventing MS compared to taking scopolamine alone at a dose of 1 mg in terms of tolerance to this syndrome. Scopolamine is used to mitigate MS, whereas dextroamphetamine is used to reduce the soporific action of the scopolamine [112,115]. The side effects, such as dizziness, dry mouth, blurred vision, anxiety, and drowsiness, make this class of drugs undesirable for long-term usage [57,58]. Dextroamphetamine is not commonly used to treat MS; therefore, there are only a limited number of research publications about this medication in treating MS. Therefore, additional research is needed to gain a better understanding of its mechanism of action.

#### 2.1.4. Corticosteroids

Glucocorticoids class of drugs, such as dexamethasone, are steroid hormones that bind to the glucocorticoid receptors [116] and can reduce nausea and vomiting through inflammation suppression and prostaglandin inhibition [117]. Dexamethasone works by inhibiting the release of certain cytokines and other inflammatory mediators [116]. It binds to glucocorticoid receptors in the cytoplasm, which then enter the nucleus and bind to specific DNA sequences, thereby suppressing the expression of genes that are involved in the inflammatory response [118]. In addition to its anti-inflammatory effects, dexamethasone also can suppress the hypothalamic–pituitary–adrenal axis, leading to a decrease in cortisol secretion [59]. It can also affect the release of certain neurotransmitters, such as serotonin [119]. Therefore, the mechanism of action of dexamethasone is complex and involves multiple pathways that contribute to its ability to modulate the immune and endocrine systems, as well as its effects on the brain and other tissues. The application of dexamethasone for the treatment of MS has been suggested based on longer-lasting effects compared to scopolamine and d-amphetamine [120]. According to the authors [59], dexamethasone decreased MS and increased endocannabinoid system levels, which were diminished after MS induction. They suggested that dexamethasone can enhance the activity of the endocannabinoid system by elevating the levels of two endocannabinoids, anandamide and 2-arachidonoylglycerol (2-AG), in the brain. This in turn leads to the activation of cannabinoid receptors in the brain, which have been implicated in the regulation of nausea and vomiting. Similar to other existing medications, many side effects such as delayed wound healing [60], high blood pressure and hyperglycemia [61], muscular weakness, and gastrointestinal disorders [60] have increased concerns about its function as a MS treatment.

In conclusion, differences in pharmacokinetics, such as how quickly the medication is absorbed, distributed, metabolized, and eliminated by the body, can also impact how effective the medication is, how long it lasts, and what side effects it may cause [121,122]. Therefore, even medications that share a common mechanism of action may still have different side effects, which should be considered when prescribing or using them.

### 2.2. Non-Pharmacological Interventions for Mitigating Motion Sickness

#### 2.2.1. Effects of Nutrition on Motion Sickness Syndrome

It has been documented that the residents of the Samoa islets consumed one or two mangoes prior to their sailing trips to prevent sea sickness [123]. This could be attributed to the high levels of polyphenols [124] and ascorbic acid (vitamin C) in mangoes [124,125]. Researchers in [123] conducted a study to assess the effects of taking oral vitamin C on seasickness symptoms. One group of participants received 500 mg of vitamin C, and another group received a placebo before exposure to motion stimuli (a lifeboat exposed to 1-m-high waves in an indoor pool) [123]. In both groups, one hour before and again 20 min after the experiment, blood samples were taken to identify different blood metabolites after vitamin C intake, compared to placebo. Their results showed that only 34% of participants reported seasickness after vitamin C intake; however, 65.07% of participants were sick in the placebo group. Therefore, they concluded that vitamin C could be considered an effective nutritional factor in reducing seasickness [123]. In another survey about vitamin B12 deficiency and susceptibility to MS, a study [126] found that there was no relationship between vitamin B12 deficiency and susceptibility to MS. Therefore, various nutrients can have different impacts on MS [5].

Based on [127,128], researchers suggested that dietary modifications and nutritional interventions may be a useful strategy for managing symptoms associated with MS. Their study showed that consumption of a meal with a combination of macronutrients including 10% protein, 30% fat, and 60% carbohydrates led to an increase in parasympathetic cardiac tone and a decrease in MS symptoms [127]. However, the effect of each macronutrient on MS was not evaluated independently; therefore, it is not exactly clear which macronutrient was effective in mitigating MS.

Lindseth et al. [5], in a recall questionnaire of 57 pilots (49 males and 8 females, 18–36 years old) 24 h prior to a flight, found that people who consumed meals high in sodium, thiamine, or calcium reported a higher risk of airsickness. According to their findings, consuming a meal that is high in carbohydrates resulted in a reduction in the symptoms of airsickness. On the other hand, Levine et al. [4] investigated the effect of consuming protein-enriched-meal (53% protein, 12% carbohydrate, and 35% fat) and carbohydrate-enriched-meal (100% carbohydrate) on MS induced by a rotating optokinetic drum on 18 participants (15 females, 3 males, 18–20 years old). The obtained results showed that meals consisting of higher amounts of carbohydrates exacerbated both peripheral symptoms (headache, sweating, and warmth) and central symptoms (drowsiness and dizziness) of MS compared to meals consisting of higher amounts of protein [4]. Differences in study design, sample characteristics, and environmental conditions could explain the variations between the two studies. The effects observed may be due to factors such as food quantity and composition, which were not clearly specified by the participants [128,129]. Therefore, further research is needed to establish the optimal dietary and nutrient regimens for alleviating MS symptoms and to investigate the reasons behind the inconsistent results in the literature. It can be suggested that the variation in the types of motion exposure (e.g., air, sea, or virtual), exposure duration, and control groups employed could potentially account for different outcomes.

Furthermore, it has been revealed that high-fat meals prior to flight resulted in increased levels of air sickness [5]. The authors in [130] examined the effect of both low-fat meals (50 g lean beef + water) and high-fat meals (30 g water + 30 g margarine) on twelve male participants who were exposed to drum rotation after the intake of meals. They reported that MS in the high-fat meal group was significantly increased compared to the low-fat meal group [130]. However, as the efficiency of protein in reducing MS has been extensively studied in [4], the amount of protein in a low-fat meal should be counted and assessed as it is missed in [130]. Foods high in fat take longer to digest, so the presence of fat in a meal can slow down gastric emptying [131]. This is because fat stimulates the release of hormones that signal the stomach to empty more slowly [132]. Research has suggested that there may be a link between gastric emptying and MS. Specifically, delayed gastric emptying has been found to be a risk factor for MS in some individuals [133,134]. Fat has been shown to delay gastric emptying as well. Researchers in [135] evaluated the effect of MS on gastric emptying. During the experiment, participants were subjected to head movements while seated in a rotating chair to induce MS. Nuclear medicine techniques were used to determine the gastric emptying of a liquid (300 mL). They observed that during the peak of MS symptoms, gastric emptying was significantly inhibited. However, gastric emptying returned to normal 15 min later when the symptoms subsided. Studies have shown that high-fat meals can lead to a slower rate of gastric emptying and an increased risk of nausea and vomiting during MS. It has been found that fat-induced ileal brake is a physiological response that occurs in the ileum, the final segment of the small intestine, in response to the presence of dietary fat [136]. When fat enters the ileum, it triggers the release of certain hormones, such as peptide YY and glucagon-like peptide-1, which can reduce appetite and slow down gastric emptying [137]. This effect can contribute to the delay in gastric emptying that occurs with high-fat meals and may increase the risk of nausea and vomiting during MS in some individuals [137].

Based on the findings of researchers in [4,5,128,130,138], nutritional interventions and dietary modifications represent a promising approach for managing symptoms associated with MS. Factors such as food composition, varying amounts of macro- and micronutrients, and meal palatability could be examined in future research studies to gain a more comprehensive understanding of the role of diet in the management of MS. By investigating these factors, researchers may be able to develop personalized and targeted dietary approaches that are highly effective in managing the symptoms of MS.

##### Plant-Based Interventions

Zingiber officinale Roscoe (ginger), a traditional medicinal plant with high amounts of bioactive compounds ‘gingerols,’ has been widely used to mitigate MS [139]. Lien et al. [140] revealed that the normal gastric slow wave (3 cycles per minute, cpm) before exposure to motion stimuli became fluctuated after rotation, which resulted in amplified gastric signal activity (4.5–9 cpm) and nausea development. However, with ginger pre-treatment (2 g), the normal gastric slow wave remained constant after rotation. The efficiency of ginger in MS studies was controversial, and different results about the anti-MS properties of ginger have been related to different amounts of this compound [141]. Researchers [142] showed that capsulated ginger powder (1 g) had no significant effect on gastric signals of MS, while other investigators [143] found that consumption of ginger (1 g) was effective in revealing vomiting, vertigo, nausea, and cold sweating symptoms of MS compared to a placebo. A survey that reviewed the effect of consuming ginger on gastrointestinal disorders stated that approximately 1.5 g ginger should be used for relieving nausea symptoms [144]. Individuals who were treated with ginger had a prolonged onset time of nausea and showed a shorter recovery time after exposure to motion stimuli [140]. Research conducted by Zaghlool et al. [145] revealed that ginger exerted its anti-MS effect through its antihistamine activity. In addition, the combination of ginger with other plants has successfully resulted in mitigating MS [139,146]. As suggested by researchers in [140], ginger could potentially reduce MS-related nausea by inhibiting gastric dysrhythmias and the increase in plasma vasopressin. Ginger’s ability to regulate sugar metabolism, control fatty acid oxidation and decrease the release of histamine and acetylcholine in the vestibular system may be the underlying mechanism for its beneficial effect on MS-related symptoms [147].

Researchers in [146] found that the herbal dietary formulation ‘’Tamzin’’ composed of ginger and tamarind, showed anti-MS properties. Tamzin might have demonstrated its anti-MS effect by modulating the activity of certain neurotransmitters in the central nervous system, including histamine and gamma-aminobutyric acid (GABA), while decreasing the activity of the excitatory neurotransmitter, such as glutamate. This modulation could have led to the suppression of the central vestibulo-autonomic pathways, resulting in relief from MS symptoms [146]. Furthermore, The high amount of vitamin C (21 mg/100 g) content in Tamzin could be relatively contributed to its antihistamine activity [123,146]. The anti-MS activity of Tamzin was also comparatively similar to that of scopolamine [146]. Similarly, the Tianxiang formulation, a combination of ginger, mint, cinnamon, and citrus family plants, has been traditionally used for the treatment and inhibition of MS in China [139]. In a study conducted by Zhang et al. [139], thirty-six rats were divided into various groups, including a control (no-rotation, no-treatment), model group (rotation, no-treatment), positive group (rotation with scopolamine) and Tianxiang group (rotation with 1.82 and 3.64 g/kg of Tianxiang treatment). All groups excluding the control group, were rotated for two hours in an acceleration stimulator [139]. The histamine and acetylcholine contents in the model group’s vestibular nucleus were higher than control group significantly, and the amounts of histamine and acetylcholine in the vestibular nucleus were less in the Tianxiang groups than control group, which declared the inhibitory effect of Tianxiang on MS [139]. The inhibitory effect of medium and high dosages of Tianxiang (1.82 and 3.64 g/kg bw per day, respectively) was more effective than scopolamine (1 mg/kg bw per day) for mitigating MS [139]. Furthermore, the anti-MS activity of Tianxiang could be correlated to bioactive components such as gingerol, hesperidin, and menthol [139]. A study by Maheswari et al. [148] showed that *Mentha arvensis* L. (mint family plant) and its potential bioactive compound, menthol, contained the highest dopamine (one of neurotransmitters involve in MS occurrence) secretion blockage among other ingredients. Menthol and mint family plants have been identified as potential antiemetic agents [149,150,151]; however, there has not been any clinical trial conducted to date that specifically examines the effect of mint and menthol on patients affected by MS [152]. It has been suggested that the possible mechanism by which menthol can affect MS is related to its cooling effect via TRP melastatin 8 (TRPM8) or cold and menthol receptor 1 (CMR1) [153]. TRPM8 is expressed in cells in the vestibular system that are involved in detecting motion, and it has been proposed that TRPM8 may play a role in modulating the activity of these cells and could potentially be targeted to treat MS [153].

Studies have demonstrated that the inhibitory effects of hesperidin on MS are linked to its antihistamine properties [148,154]. Hesperidin has been found to decrease histamine levels and downregulates histamine H_1_ receptor protein expression and mRNA in the hypothalamus and brainstem [154,155]. Researchers in [156] also found that the antihistamine activity of hesperidin was almost two times more than that of menthol.

A research study by Deshetty et al. [154] investigated the effects of pre-treatment with dimenhydrinate or hesperidin on MS levels. The study found that both dimenhydrinate (20 mg/kg bw) and hesperidin (80 mg/kg bw) significantly reduced MS levels. The antihistamine activity of both substances was suggested to underlie their observed effects. Furthermore, no significant difference was observed between the effects of the two substances at their respective doses. Based on these findings, both dimenhydrinate and hesperidin may have potential as therapeutic agents for MS.

Therefore, it is hypothesized that the observed reduction in MS levels resulting from the administration of the Tianxiang and Tamzin herbal formulations is attributed to the potential presence of bioactive compounds, including hesperidin, menthol, and gingerol, which possess antihistamine properties. These compounds may exert their effects by modulating histamine levels in brainstem regions and the hypothalamus. Based on the evidence presented in [148,154]. However, further research is needed to confirm the effectiveness and safety of these herbal formulations for MS treatment. Table 2 shows detailed information about various nutritional interventions on MS.

#### 2.2.2. Effects of Music on Motion Sickness

Methods that stimulate parasympathetic activity may be helpful in alleviating MS, as a decrease in parasympathetic activity has been associated with an increase in susceptibility to MS. Therefore, it can be concluded that increasing parasympathetic activity could help reduce the severity of symptoms [127,159].

Music has been widely studied in various contexts to help reduce stress and promote relaxation [160]. There is some evidence to suggest that this may be effective in reducing MS symptoms as well [161].

Researchers evaluated the effects of music on autonomic nervous system activity, as well as the impact of favorite music on parasympathetic nervous system activity after exercise sessions. The study comprised twenty-six subjects who were divided into two groups: one group that listened to their favorite music and another that did not listen to any music. The results of this study showed that heart rate variability, which is an indicator of parasympathetic nervous system activity, was enhanced after listening to favorite music. This suggests that listening to music may have a positive impact on the parasympathetic nervous system and help promote relaxation [161].

Keshavarz and Hecht [162] examined how pleasant music affects VIMS. They exposed 93 participants to a 14 min virtual bicycle ride while playing various types of music such as neutral, relaxing, stressful, desirable music, or no music (control group). The researchers found that listening to desirable music, which was music that participants had previously rated as enjoyable, significantly reduced VIMS compared to the other music groups and the control group. The authors suggest that pleasant music may have a positive impact on the parasympathetic nervous system, which could help alleviate the severity of VIMS symptoms. However, it is important to note that this study had a relatively small sample size, and further research is necessary to better understand the relationship between music and VIMS [162].

Additionally, the effects of music on MS were investigated by researchers in [161]. The study involved 24 participants who were exposed to cross-coupled Coriolis stimulation, which can induce MS by causing a misperception of body orientation. The participants experienced mild nausea, and then a specific piece of music called “Travelwell” was played for them. The motion continued until they experienced moderate nausea. The results indicated that the time for mild nausea to reach moderate nausea increased significantly from 9.2 ± 5.9 min for the control group (without music) to 10.4 ± 5.6 min for the music group. This suggests that listening to Travelwell music may have a protective effect against MS. However, one limitation of the study is that the Travelwell music used was specifically produced to reduce MS, and it is unclear how other types of music may compare in their effects.

Similarly, another study evaluated the impact of valence, arousal, and likability of music on MS symptoms. The study included 80 participants (63 females, 17 males) with a mean age of 26.95 years, who were asked to watch a bicycle riding video. Out of the total participants, 40 (32 females, 8 males) were randomly exposed to different types of classical music (happy, peaceful, agitated, and sad) while watching the video. In order to assess the likability of music on MS symptoms, participants were divided into two groups: those who liked the pre-selected music (n = 21) and those who disliked it (n = 19). An additional 40 participants (31 females, 9 males), with a mean age of 24.90 years, were randomly assigned to either a group that listened to their self-selected favorite music (n = 20) or a control group with no music (n = 20). The results showed that the participants who listened to their favorite music had a significant decrease in MS symptoms compared to the control group who did not listen to any music. These findings suggest that music can have a positive impact on MS symptoms and could potentially be used as a complementary therapy [163].

Therefore, it can be suggested that music may have a positive impact on the parasympathetic nervous system, which could help reduce the severity of MS symptoms. Specifically, listening to pleasant music that participants enjoy may increase parasympathetic nervous system activity and promote relaxation, which can alleviate MS.

#### 2.2.3. Effects of Diaphragmatic Breathing on Motion Sickness

It has been shown that one effective technique to increase parasympathetic nervous system activity is to decelerate the breathing rate [161,164,165]. The mechanism of Diaphragmatic Breathing (DB) on MS reduction is not totally understood, but it may activate the inhibitory reflex between vomiting and respiration [161]. In order to achieve the highest activation of the parasympathetic nervous system, it was claimed that DB should be at a rate of three to seven breaths per minute [166]. Researchers in [62] examined the effect of DB on 43 individuals susceptible to MS who were randomly assigned to a slowed-pace DB group at six breaths-per-minute, or normal pace breathing. Participants were exposed to a VR environment showing a boat in rough sea conditions for almost 10 min. The effect of DB on MS, respiration rate, and heart rate variability was recorded before, during, and after experiments. They found that DB had higher activation of the parasympathetic nervous system and reported fewer MS symptoms during VR application [62]. Increased DB resulted in reduced respirations per minute which demonstrated that DB could be considered as a potential approach to increase parasympathetic tone and resulted in MS reduction [62]. It has been confirmed that cortisol levels in participants with MS were significantly higher than in non-susceptible individuals [167], and the DB method resulted in lower cortisol levels in plasma [168]. In a study about the effect of DB on MS [62], sixty participants were randomized to one of the study groups, including a control group, focusing on environmental alertness, or a paced DB group. Participants experienced VIMS by watching a 10 min video of ocean swells during a stormy condition. The results have shown that using a controlled DB approach is a rapid and effective method for reducing MS [62,161].

#### 2.2.4. Effects of Odor on Motion Sickness

Among factors that relate to the worsening of MS, the presence of an unpleasant and robust odor is associated with exacerbating the symptoms of MS, especially in susceptible individuals [169]. It has been claimed that unpleasant perceived odors such as tobacco smoke and pyridine (structurally related to benzene) have been associated with increased MS [31]. Keshavarz et al. [170] reported that exposure to rose fragrance as a pleasing odor and leather as an unpleasing odor significantly decreased VIMS; therefore, they concluded that an olfactory experience might have an impact on VIMS, and MS index can be reduced with a pleasant perceived odor. On the other hand, a study conducted by Paillard et al. [169] showed no significant influence of pleasant (limonene) or unpleasant odor (petrol) during rotation on MS reduction. In a separate experiment, it was demonstrated that individuals who are highly sensitive to unpleasant odors and have a preference for sweet flavors are more susceptible to developing MS [171]

#### 2.2.5. Effects of Administration of Fresh Air on Motion Sickness

Long-distance travel in an overloaded vehicle without an efficient ventilation system increases the chance of exposure to foul air and the possibility of unpleasant feelings, discomfort, headaches, and sweating (cold) symptoms of MS [172]. The administration of fresh airflow has been described as a successful technique to alleviate MS symptoms [129]. It has been stated that cooling the body temperature through sweating reduces body metabolism, which assists in reducing MS. Therefore, airflow may speed up this action by lowering the core body temperature through evaporating sweat [173,174]. According to Keshavarz et al. [6], the mechanism of decreasing MS can be linked to the airflow effect on temperature, promoting a pleasant atmosphere and reducing sensory conflict. In research conducted by D’Amour et al. [175], one group of participants was exposed to the airflow generated from two fixed fans positioned near the bicycle video screen, and in a control group without airflow conditions. The VIMS in the airflow group decreased significantly compared to the control group. Furthermore, while airflow decreased the nausea and oculomotor features of VIMS, no significant impact of airflow on symptoms associated with disorientation was found [175]. Therefore, it has been suggested that providing airflow can be considered an operative, affordable, and simple way to mitigate VIMS [6].

#### 2.2.6. Effects of Autogenic-Feedback Training on Motion Sickness

Autogenic-feedback training (AFT) has been utilized in space programs as an intervention for mitigating MS and was developed for training astronauts to control their physiological responses before long-duration space flights [176,177]. AFT is a combination of several perceptual and physiological training methods that incorporate the progressive relaxation method, autogenic psychotherapy, and biofeedback techniques [178]. Among its uses, AFT has been used to improve the performance of pilots and astronauts, reduce airsickness caused by high-performance military aircraft, train the cardiovascular and other autonomic responses of MS patients, as well as treat and help with many other debilitating diseases [85]. In a study [176] by Acromite et al., 20 subjects (24–65 years old) participated in either an AFT program or no treatment group (control), and a cross-coupled rotation chair was utilized to induce MS with a speed of six revolutions per minute (rpm) which was constantly maintained for 5 min and raised by 2 rpm at 5 min intervals until the tests were completed, or participants reported severe nausea. Their study found that subjects undergoing AFT maintained their heart rate and skin conductance at lower levels in comparison to controls [178]. In order to compare the anti-MS efficiency of AFT and pharmacologic treatment (promethazine), a rotating chair for thirty minutes (four days per week for three weeks in a row) was applied to the AFT group. Their outcomes showed a lower incidence of MS in the AFT group compared to the promethazine group, and the AFT program increased the onset of MS about 2–3 times more than the promethazine condition [86]. This AFT method was based on decreasing the activity of the sympathetic nerve system [178]. No side effects have been identified for AFT exercise [179].

#### 2.2.7. Effects of Habituation Program on Motion Sickness

Habituation is a dynamic central nervous reaction induced by repeated exposure to a specific motion [115]. The habituation program has been considered one of the most effective techniques, which has been used in many countries such as Canada, the United Kingdom (UK), and Italy to diminish MS susceptibility [171]. It has been found that the rate of habituation was much slower when the subjects were treated with medicines [135]. For instance, scopolamine may postpone habituation either indirectly or directly through sedative effects [135]. It has been revealed that the direct association between age and MS is because of habituation to frequent exposures [180]. People who frequently participate in a specific kind of motion-related stimulus, virtual environment, or motion vehicle might experience adaptation or habituation to MS [181], as Palmisano et al. found that a habituation process might be contributing to the VIMS reduction when people were exposed to the same VR game, repeatedly [182]. There was a significant reduction in MS nausea symptoms on the last day of an experiment following five consecutive days of using VR via a Head-Mounted Display, and the researchers concluded it could be due to habituation [183]. Therefore, they suggested that this may have implications for the design of VR applications and systems. Specifically, they indicated that ongoing exposure to VR environments, with breaks to prevent overload, may assist users to habituate to the side effects of VIMS and improve their experience of presence in the virtual environment [183]. Repetitive rotation for 21 days resulted in a notable upregulation of the expression of Aquaporin-1 protein (involved in the transmembrane passage of water molecules) in the inner ear by 259% [184]. It has been suggested that the Aquaporin-1 protein could be an endogenic MS sensitivity regulator, and it is connected to the habituation process of MS [184]. A study conducted by Cowings and Toscano [86] showed that 30–40% of neophyte pilots experienced airsickness on their first air flight, but most of them adjusted after their third or fourth flight experiences. Similarly, a study of four consecutive trips was conducted by Nunes et al. [157], and a significant reduction in total MS score was observed from the second trip to the fourth trip, compared to the first trip. This confirmed the habituation occurrence during repeated trips. Habituation is preferred to anti-MS drugs by the military service’s programs as it is free of side effects, whereas anti-MS medicine is contraindicated for pilots due to their unwanted side effects, such as blurred vision and drowsiness [86]. The success rate of habituation programs exceeds 85% [185]; however, they can be extremely time-consuming and costly [185,186]. These above-mentioned studies suggest that behavioral countermeasures, such as listening to favorite music, the effect of DB, favorite odor, administration of fresh air, application of an anti-sickness program (AFT), and habituation possess a significant potential to reduce MS, and they can be applied to lessen other subcategories of MS, such as simulator sickness, and cybersickness (see Figure 4).

## 3. Discussion

MS, due to the conflict between signals from the vestibular and visual systems to the brain, leads to undesirable symptoms. This has been the focus of researchers for decades worldwide. Despite existing medications being prescribed widely for MS mitigation in a real environment, such as a car, boat, plane, or train, they have not been used in VR and simulators due to potential side effects. Therefore, research efforts have been increased recently toward identifying appropriate approaches to alleviate MS without the side effects of currently available medications [187,188].

It has been shown that the efficiency of existing medications such as cyclizine, dimenhydrinate, meclizine, promethazine, and cinnarizine is because of their antihistamine activity [189]. At an equivalent dose (50 mg), cyclizine showed slight anti-MS activity over dimenhydrinate and had fewer side effects. For these reasons, cyclizine has been recommended for children traveling [190]. Researchers found that meclizine had a slower start point with a longer duration of action (12 to 24 h), but it was associated with a wider range of side effects [190]. It has been indicated that scopolamine is more effective for MS treatment in comparison to cinnarizine [91,108]. Scopolamine prevents vestibular signals to the vestibular nuclei and possibly has a direct effect on the vomiting center in the brain [190]. Scopolamine averts connections of the vestibule nerves to the vomiting center by hindering the acetylcholine neurotransmitter action [91,190]. On the other hand, there is no information to indicate any advantages of dextroamphetamine compared to scopolamine or antihistamine medications. However, they are often taken incorporated with scopolamine or antihistamine medications to decrease their side effects [189,190]. A research study by Wood et al. [68] showed that meclizine, dextroamphetamine, and scopolamine and their combination delayed the onset of MS by 50%, 70%, 147%, and 194%, respectively. They concluded that the single effective drug against MS was 0.6 mg of scopolamine; however, amplifying the dose of scopolamine to 1.2 mg was not effective in the prevention of MS. In addition, the doubled dose of scopolamine increased its side effects [33]. However, the common side effects of scopolamine such as dry mouth, drowsiness [54], impaired memory [55], blurred vision, headache, nausea, abdominal pain, dizziness, sweating [31], tachycardia, urinary retention, and acute angle-closure glaucoma [56] (see Table 1) have resulted in the search for other possible treatments such as behavioral interventions or nutritional approaches to mitigate MS.

Nutritional and behavioral countermeasures have been compared to pharmacological interventions in the past. For instance, researchers in [191] evaluated the efficiency of ginger and dimenhydrinate in thirty-six participants (18 f, 18 m aged between 18–20 years) who were exposed to six minutes of motor-driven revolving chair simulator after taking either capsulated ginger (940 mg) or dimenhydrinate (100 mg). According to the study, half of the ginger-treated participants tolerated the full rotation sessions, whereas none of the dimenhydrinate-treated subjects were able to tolerate the entire 6 min rotation duration; therefore, ginger is considered to be an effective anti-MS agent [192]. Researchers have shown that ginger can exhibit varying levels of efficiency depending on the extraction factors of 6-gingerol and its combination with other plant substances, which can have a direct impact on MS studies and results [141]. A biochemical study demonstrated that ginger could alleviate MS by decreasing histamine and acetylcholine release, regulating sugar metabolism and fatty acid oxidation, and reducing MS by 53.7% when compared to a non-ginger group [147].

It has been stated that subjects who used the AFT program tolerated all types of MS experiments, whereas control subjects (without using the AFT program) did not [187]. A study revealed that DB had a significantly greater effect on MS symptoms than music; however, in their research, they only evaluated one specific audiotape, not all types of music [161]. As a result, this can lead to different consequences in MS studies [162]. Other investigators [6] reported that pleasant music, regardless of the music type, has the potential to reduce VIMS compared to the unfavorite or no-music groups [162]. It has been suggested that pleasant conditions which result in comfortable and enjoyable perception might have the potential to reduce or inhibit MS [6].

Several studies have examined the impact of protein and carbohydrate consumption on the autonomic nervous system [4,127]; however, the relationship between macronutrients and MS is intricate and may be influenced by multiple factors, such as dietary composition and structure [4,128,193]. While some studies have suggested that protein intake can alleviate MS symptoms [4], others have reported conflicting results [5]. This may be partly explained by the influence of other macronutrients, including dietary fat content [194], on protein digestion and absorption. The kinetics of protein hydrolysis may also affect amino acid absorption and digestibility [193], which may, in turn, impact the effectiveness of protein in reducing MS symptoms. High-fat diets have been linked to higher MS symptoms [130], possibly because of their negative impact on the availability of food components, including protein [193]. Therefore, when investigating the relationship between protein and MS, it is important to consider the impact of dietary composition, including macronutrient ratios and other factors that can modify amino acid absorption and availability. Based on a review of the literature, the composition of pre-exposure meals appears to significantly impact the severity of MS symptoms. While the effects of high-carbohydrate and high-protein meals have yielded varying results, the consumption of high-fat meals prior to exposure to MS has been associated with a heightened risk of this syndrome. Further research is necessary to elucidate the macronutrients or ingredients that most significantly influence MS symptoms.

The inhibitory effects of plants and their bioactive compounds, such as *Mentha arvensis*, hesperidin, and menthol, on MS are related to lowering histamine levels. The anti-MS effect of plant-based nutrient substances such as Tianxiang and Tamazin was similar to scopolamine, and hesperidin was similar to dimenhydrinate medicine. Studies involving ginger were inconclusive due to varying ginger doses between trials and study designs. According to Table 2, nutrient substances such as ginger [140], Tamazin [146], Tianxiang formulation [139], menthol [156], hesperidin [154], and vitamin C [123] can reduce MS through lowering histamine [139,146,148,154], dopamine [148,156], and acetylcholine levels [139]. There may be additional mechanisms that contribute to the effectiveness of these substances, but further evaluations are necessary to confirm their impact.

### 3.1. Limitations

After reviewing current studies on MS, we noticed several significant limitations that should be carefully considered. In the following, the most significant limitations are listed.

#### 3.1.1. Data Collection Impact

One of the main challenges for MS detection/reduction is the data size. In other words, our current review reveals that most previous studies had a limited number of participants in their experiments. This point may impact the reliability of the obtained outcomes. Even though having an adequate number of participants is not always easy to achieve, we believe that this is a serious limitation of previous studies that need attention in the future.

We also noticed that the impact of critical factors, including different macro and micronutrients and different bioactive compounds, were not studied properly in MS studies. It should be mentioned that the impact of these factors on various medical diseases has been proven.

Furthermore, we noted that in most studies, factors such as age and gender were not equally distributed, as they can have an impact on MS. For instance, it has been shown that men and women have different levels of MS in the same environment [195]. Thus, their investigation is highly recommended. Another crucial limitation of previous studies is that the datasets are mostly imbalanced. As a result, the collected datasets were not balanced, which had a significant impact on the analysis. In the study of Howard and Van Zandt [196], the authors found that women experienced more VR sickness than men. This is a significant point in terms of gender considerations during the data collection process.

#### 3.1.2. Neuronal Mechanism of MS

Based on the sensory conflict theory, MS results when the brain perceives conflicting signals regarding body movements from the visual and vestibular receptors and the proprioceptive system. There is, therefore, a close connection between neuronal mechanisms and MS severity [29]. While there are several studies that examine neuronal mechanisms and MS, we believe a deeper analysis of the underlying reasons for MS and VIMS in the brain should be conducted [26].

#### 3.1.3. Clinical Intervention

There have been a lot of studies that have utilized animal models, such as rats and mice, to study MS. However, it is important to keep in mind that these animals lack the ability to vomit and have a different perception of nausea compared to humans [197]. Therefore, it is crucial to take into consideration the mechanisms behind their responses and to conduct clinical studies in addition to animal model studies [198].

### 3.2. Research Gaps and Future Directions

Even though up to now, a remarkable effort has been made to detect and reduce MS, our review reveals several key research gaps and future directions. In the following, we discuss some of the most significant research gaps in the domain that should be further investigated.
Most complementary and alternative treatments have not been tested in clinical trials, and research is needed to determine their impact on human physiological and biochemical metabolites during MS occurrence.Based on our literature review, a significant gap in the existing research has been identified with respect to the potential influence of taste, food texture, and flavor on MS levels. Despite the critical role that these factors play in determining our dietary choices, their impact on MS levels has not been investigated previously. Therefore, there is a pressing need for further rigorous research to investigate the potential role of taste, food texture, and flavor in modulating MS levels.It is important to note that the impact of nutrition may vary based on multiple factors, including the person’s overall diet and health, as well as the specific type and amount of protein and carbohydrates consumed. Further research is necessary to fully comprehend the connection between protein, carbohydrates, and MS.As physiological signals are affected by MS, a comprehensive study of physiological responses to MS in both real and virtual environments would be a valuable contribution to our understanding of this phenomenon. It may help develop effective preventive and therapeutic strategies.Along with the effect of music and DB, the effect of distraction should be investigated extensively. Lin et al. [199] distinguished four main challenges to driving cognitive research, including distraction, drowsiness, navigation, and MS. Based on their recommendation, these four main challenges are key to studying brain activity in different experimental paradigms.Some research studies have suggested that a combination of multiple techniques could more successfully minimize MS symptoms. Therefore, future studies should consider attempting to combine methods such as the effect of both favorite music, DB, and favorite odor on MS.Given that the impact of vestibular deficiency has been studied in real-world environments, it is worth investigating in future research whether vestibular system impairment can affect the severity of VIMS.The impact of artificial intelligence (AI) technology (machine learning (ML) and deep learning (DL) as well-known AI technologies) is a gap in MS research that should be investigated more comprehensively. Several studies have been conducted on the use of traditional ML and DL methods for MS. Based on the results, we found that traditional ML techniques are most frequently used compared to DL methods. As we discussed limitations in data collection above, DL methods require huge amounts of data to provide meaningful results. Therefore, we highly recommend applying different ML and DL approaches to find hidden patterns in MS data as well as providing an automated detection/reduction framework. On the other hand, our findings show that there is no study on uncertainty quantification (UQ) of ML and DL methods. Even though ML and DL can provide excellent results, the issue of dealing with their uncertainty remains unresolved. In Table 1, we list and discuss the most relevant studies on MS with the application of ML and DL methods.

According to Table 3, it can be observed that there are very few studies focusing on the application of ML and DL methods in MS. Moreover, we can see that almost none of the previous studies on ML and DL methods in MS did consider the uncertainty of their proposed models. Based on these findings, we highly recommend developing more reliable AI-based MS detection/reduction systems.

## 4. Conclusions

In this paper, we reviewed the most recent advanced approaches towards MS reduction by having less medication. According to the current literature review, methods that activate the parasympathetic nervous system, such as diaphragmatic breathing, listening to favorite music, exposure to fresh air, and pleasant odors, have been used to reduce MS in both VR and real environments. Further, the research found that nutrients and bioactive compounds such as Vitamin C, ginger, hesperidin, and menthol have successfully reduced MS. It can be concluded that the concept of behavior approaches along with nutrition consumption provides a new foundation to manage MS symptoms. This foundation will enable the elimination of unwanted side effects from existing medications. Thus, this can collectively improve society’s health and provide safe, affordable, and convenient ways to mitigate this undesirable syndrome while traveling by car, ship, plane, and train in a real-world environment as well as while using simulators and VR applications. We also listed the significant research gaps and future research directions for investigators who would like to work on MS.

## Figures and Tables

**Figure 1 nutrients-15-01320-f001:**
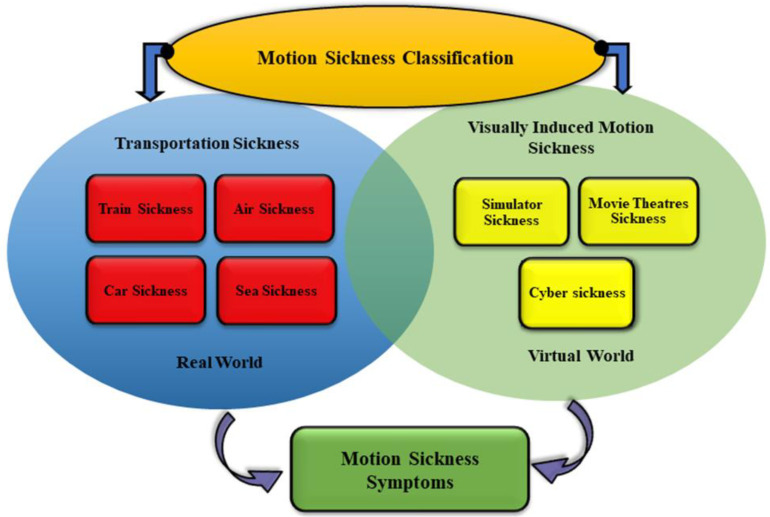
Classification of MS in both real and virtual environments. Airsickness, seasickness, carsickness, and trainsickness can occur on airplanes, ships, cars, and trains, respectively, while cybersickness, simulator sickness, and movie theater sickness can be caused using head-mounted displays, simulators, video games, and large 3D screens, respectively. Despite MS has been categorized into different subsections; however, the main symptoms are the same.

**Figure 2 nutrients-15-01320-f002:**
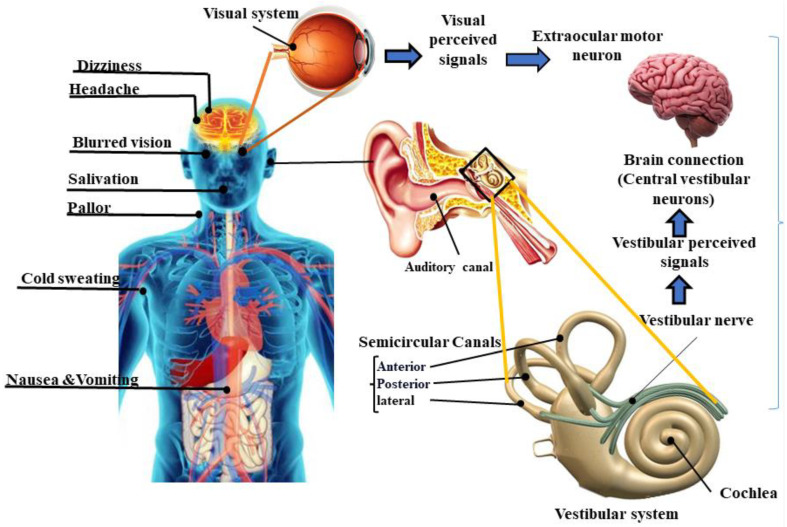
The vestibular system within the inner ear, transfers signals and information about perceived motion, spatial orientation, and head position to the brain. On the other hand, if the brain perceives and provides different inputs to the visual system, therefore, unauthentic inputs from these organs become a stimulus to trigger symptoms of MS.

**Figure 3 nutrients-15-01320-f003:**
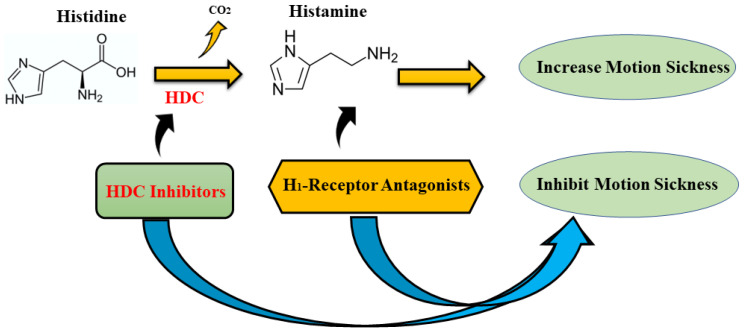
Histamine is synthesized from the amino acid histidine via histidine decarboxylase (HDC) enzyme [74]. MS can be successfully inhibited by the effect of HDC inhibitors, or/and first-generation histamine H1-receptor antagonists (H_1_-receptor antagonists).

**Figure 4 nutrients-15-01320-f004:**
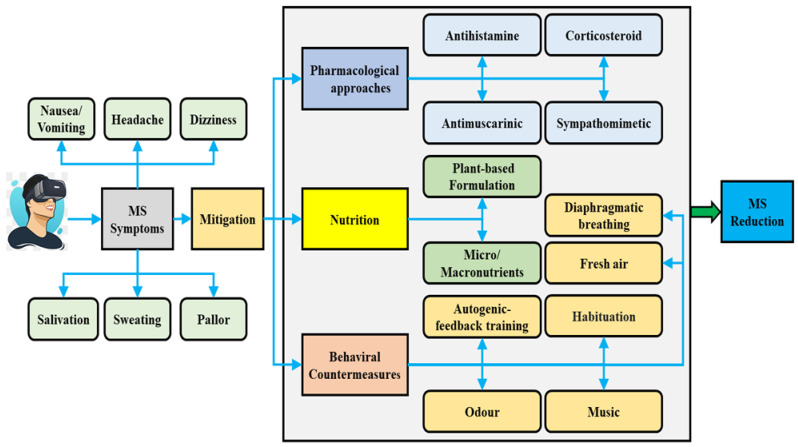
Motion Sickness (MS) can lead to similar symptoms in both real and virtual environments. Multidisciplinary interventions used for MS include pharmacological approaches, nutritional interventions, and behavioral countermeasures such as favorite music, autogenic-feedback training, fresh air, odor, diaphragmatic breathing, and habituation.

**Table 1 nutrients-15-01320-t001:** Common pharmacological treatments in inhibiting Motion Sickness.

Name	Protection Index (%)	Mechanism	Used Dosage	Side Effects	References
dimenhydrinate	72.91	Histamine H_1_ receptors antagonist	25 mg to 50 mg 32 (mg/kg)	Headache, drowsiness, blurred vision, eye irritation, drowsy, dizziness, concentration difficulty, fatigue, euphoria, and hallucinations	[30,33,34,35,36]
betahistine	weak effect	H_3_ presynaptic antagonist and a partial H_1_ postsynaptic agonist	8 mg–32 mg	Nausea and vomiting, gastric upset and decreased appetite	[28,32,37,38]
α-fluoromethylhistidine (α-FMH),	Not mentioned	Histidine decarboxylase inhibitor	100–200 mg/kg	Sedation effect	[39,40]
cinnarizine	Not mentioned	Histamine H_1_ antagonist	50 mg	Drowsiness, dry mouth, sedation and, Parkinson’s disease	[30,41,42,43]
promethazine	78 64.33	Histamine H_1_ receptors antagonist	25 to 50 mg	Drowsiness—akathisia—Restlessness dry mouth and extrasystoles	[32,35,44,45,46]
cyclizine	71.24	Histamine H_1_ receptors antagonist	50 mg 3 times a day	Dry mouth, blurred vision, and drowsiness	[33,47,48,49]
meclizine	67.98	Histamine H_1_ receptors antagonist.	20 mg/kg	Drowsiness, dry mouth, and constipation	[33,35,50,51]
scopolamine	78 62.96	Acetylcholine antagonist	0.6–1.2 mg for adults 0.25 mg for children	Dry mouth and drowsiness Reduced memoryblurred vision, headache, nausea, abdominal pain, dizziness, sweating, tachycardia, urinary retention, and acute angle-closure glaucoma.	[31,33,35,52,53,54,55,56]
dextroamphetamine	64	Not well-described	5 to 10 mg in combination with other antihistamines	Dizziness, dry mouth, blurred vision, anxiety, drowsiness, and the risk of drug dependence	[29,33,57,58]
dexamethasone	Not mentioned	Increased endocannabinoids	0.05 mg/kg	Delayed wound healing, high blood pressure, hyperglycemia muscular weakness, and gastrointestinal disorders	[59,60,61]

**Table 2 nutrients-15-01320-t002:** Effect of different food consumption in a real and virtual environment and their impact on MS.

Food Type	Number of Participants Age and Sex	Study Design	Used Dosage	Apparatus	Mechanism	Effect on MS	References
Ginger	N = 64: 32 females, 32 males 20–38 years	balanced placebo design’	1000 mg	rotating vection drum	not identified	unchanged	[142]
	N = 184: 18–65 years	open single-arm study	160 mg	trip for 15 min	not identified	unchanged	[157]
	N = 18: 8 males and 10 females; 18–40 years	double-blind, randomized, placebo-controlled study	1000 mg 2000 mg	circular vection	prevent the release of vasopressin	reduced nausea	[140]
	N = 80: 16–19 years	double-blind randomized design	1000 mg	full-rigged training-ship	Not identified	unchanged	[143]
Tamazin	N = 36 mice	randomized trial	400 mg/kg.bw 800 mg/kg.bw 1200 mg/kg.bw	self-manufactured rotation device	reduced histamine level	reduced	[146]
Tianxiang	N = 60: male rats	randomized trial	0.91 g/kg, 1.82 g/kg and 3.64 g/kg per day,	biaxial rotating acceleration stimulator	reduced histamine and acetylcholine	reduced	[139]
Menthol	N = 36, female mice	randomized trial	50 µg/mL	custom-designed centrifuge machine	reduced dopamine	reduced	[156]
Hesperidin	N = 36 female mice	randomized trial	80 mg/kg bw	custom-designed centrifuge machine	reduced histamine and GABA	reduced	[154].
Protein	N = 18: 15 femalesand 3 males; 18 to 20 years	repeated measure design	53%protein + 12% carbohydrate and 35% fat	rotating optokinetic drum	Increased parasympathetic tone.	reduced	[4]
	N = 57: 49 males and 8 femals; 18 to 36 years	descriptive correlational study	Not mentioned	real flight	Not mentioned	increased	[5]
Carbohydrate	N = 108: 68 females and 40 males; 18–23 years	A double blind, independent subject design with modified random assignment	49.2 g carbohydrate, 4.8 g fat, 12 g protein,	rotating optokinetic drum	Not mentioned	not effective reduced.	[4,5] [128]
Fat	N = 12: 12 males; 22 to 36 years	Randomized trial	-high-fat meal: 30 g water +30 g margarine. -low fat meal: 50 g beef (1.5 g carbohydrate, 3.6 g protein) + 150 g water,	Rotating vection drum	Not mentioned	increased	[130]
Vitamin C	N = 70: 20 females +50 males; 19 to 60 years	Double-blind placebo-controlled crossover study	500 mg	Inflatable life raft exposed to 1-m-high waves.	decreased histamine level	reduced	[123]
Vitamin B12	N = not mentioned (18 years or older)	Not mentioned	1000 μg,	Rotator chair assembly	Not mentioned	not effective	[126]
Sodium	N = 57 (49 males and 8 females; 18–34 years)	Descriptive correlational study	Not identified	Real flight	Not mentioned	increased	[5]
Thiamine	N = 57: 49 males and 8 females; 18–34 years	Descriptive correlational study	Not identified	Real flight	Not mentioned	increased	[5]
Calcium	N = 57: 49 males and 8 females; 18–34 years	Descriptive, correlational study	Not identified	Real flight	Not mentioned	increased	[5]
Yogurt	N = 7 (age and gender have not mentioned)	Not mentioned	Not identified	Coriolis stimulation (rotating chair)	Not mentioned	increased	[158]
Mango	Not mentioned	Observation report	2–3 mangos	Sailing trips	Not mentioned	decrease	[123].
Food composition	N = 40: 21 males and 19 females; 19.3 mean years	Repeated-measure, counterbalanced crossover design.	10% protein, 30% fat, and 60% carbohydrates	Rotating drum	increased parasympathetic tone	decreased	[127]

**Table 3 nutrients-15-01320-t003:** Application of ML and DL methods in MS.

Study	Year	Environment	Participants	ML or DL	UQ
Yu et al. [200]	2010	Realistic driving environment	N/A	ML	No
Padmanaban et al. [201]	2018	3D videos	N/A	ML	No
Hell and Argyriou [202]	2018	Rollercoaster simulation tool	23 subjects	DL	No
Li et al. [202]	2019	Car driving video	20 subjects	ML	No
Lee et al. [203]	2019	360° stereoscopic video	19 videos	DL	No
Jeong et al. [204]	2019	360° video	N/A	DL	No
Martin et al. [205]	2022	Video game sessions	103 subjects	ML	No
Keshavarz et al. [206]	2022	Video of a bicycle ride	43 subjects	ML	No
Tan et al. [67]	2022	On-road driving scenario	12 subjects	ML	No

## Data Availability

Not applicable.

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
