# Peer review of "Nutritional and Behavioral Countermeasures as Medication Approaches to Relieve Motion Sickness: A Comprehensive Review"

_nutrients, 2023, doi:10.3390/nu15061320_

Round 1

Reviewer 1 Report

The purpose of this study is to review various pharmacological and non-pharmacological interventions and their effects on reducing motion sickness. Because commonly used medicines have side effects, this article focuses on non-pharmacological intervention strategies, including nutrients, plant-based interventions, music, diaphragmatic breathing, odor, fresh air, etc. At the beginning of this article, authors also introduced that there are two types of motion sickness: real and virtual.

However, neither the first paragraph nor Figure 1 explains the difference between these two mechanisms. For example, the difference in perceptual conflict between the two conditions on the stimulus properties of the vision, vestibular system, or proprioception. The visual stimulation of virtual motion sickness is much higher than that of vestibular or proprioception; while the stimulation of real motion sickness to vestibular or proprioceptive is more complex and obvious than virtual motion sickness.

Furthermore, are individuals predisposed to real motion sickness also susceptible to virtual motion sickness? The two types of motion sickness are induced differently, and although they produce similar symptoms, do they have different physiological responses? For example, whether two types of motion sickness respond differently to heart rate variability.

Therefore, it is suggested that this review revise or supplement these viewpoints to increase the value of the article. In addition, according to the content of the text, the following suggestions are provided:

1.     At the end of the first paragraph of the introductory section: Reference 4 states that individuals without the balance organ of the inner ear were immune to motion sickness during sea voyages. Is there any literature showing that such individuals have a similar immune effect on virtual motion sickness?

2.     Effects of nutrition on MS syndrome:

I.          For reference number 91, meals high in sodium, thiamine, and calcium are associated with a higher risk of airsickness. This study also found that high protein is also associated with the risk of airsickness, and it is suggested that this information should be supplemented.

II.        The conclusions of the 18th and 91st references on the exacerbation of motion sickness symptoms caused by high-carbohydrate and high-protein foods are just the opposite. In the 91st paper, it is proposed that high-carbohydrate foods can reduce airsickness; while high-protein foods can increase motion sickness risk (should be considered for inclusion in Table 2). However, reference 18 suggested that high-carbohydrate foods were more likely to exacerbate motion sickness symptoms than high-protein foods. The author did not comment on this contradiction. Is it because the motion sickness conditions discussed in these two studies are different? One is real motion sickness; the other is virtual motion sickness.

3.     The description of Figure 4 in the text tends to make people feel that the effect of pharmacological or non-pharmacological intervention on motion sickness is aimed at reducing virtual motion sickness, while ignoring whether it still has the same effect on real motion sickness (i.e., airsickness, sea sickness). It is recommended to modify text descriptions and illustrations to avoid doubts or misunderstandings for readers.

4.     The last second paragraph of the Discussion section, related references 18 and 93, describe the parasympathetic activity of protein and carbohydrate consumption: This sentence is not easy to understand, and the evidence is biased. A reorganization is recommended. Protein and carbohydrates have been found to decrease vagal activity [18], but paradoxically, they increase parasympathetic tone [93]. The authors do not comment on this controversy. Furthermore, previous studies have also indicated inconsistent effects of protein and carbohydrates on motion sickness induced by different methods [ref. 18 vs. 91], it is therefore inappropriate for the authors to conclude that high protein and carbohydrate meals reduce motion sickness and present them in the ABSTRACT.

5.     This article reviews some behavioral strategies for coping with motion sickness. Authors may consider the following paper: Tu, Min-Yu, et al. "Effect of Standardized Yelling on Subjective Perception and Autonomic Nervous System Activity in Motion Sickness." International Journal of Environmental Research and Public Health 18.23 (2021): 12854.

Please check the above references to see if they enhance the comprehensiveness of this article.

Author Response

Dear reviewer,

We would like to express our deepest gratitude for the time and effort you took to review our manuscript. Your insights and suggestions have been extremely beneficial and permitted us to improve the quality of our paper.

We have thoroughly considered your comments in detail and have created the necessary revisions to the manuscript in response to your suggestions. We hope that the modifications address your concerns and that you will find the revised manuscript to be of a high standard.

we appreciate your expertise and the time you took to review our work. Thank you again for your time and effort. We look forward to the publication of our work.

Sincerely,

Reviewer 2 Report

This is a superb collection of the literature related to the treatment of motion sickness, but in many areas, there is a lack of critical assessment and a tendency to introduce subjective descriptive phases which hides an absence of depth of understanding. Examples of the latter are:

Section 2.1.1.

·       “suppression of vestibular system function” – what exactly does this mean?

·       “fewer sedative effects” – do you mean less sedation?

·       “shown moderate antimuscarinic activity along with its antihistamine activity” – don’t know what this means – are you talking about differences in receptor affinity/efficacy or numbers and severity of side-effects thought to be related to each type of activity?

·       “with slight anticholinergic and sedative effects” – as above

·       “through its antihistaminic and calcium channel blocker activity” – how does the latter relate to anti-emetic activity?

Section 2.1.2.

·       “Antimuscarinic agents stop the action of the muscarinic acetylcholine receptor” – there are 5 muscarinic ACh receptors and only 2 of these are likely to be involved in motion sickness; please be more specific

·       “The mechanism through which Scopolamine prevents MS is elusive. It has been identified that Scopolamine acts as an antiemetic agent mainly through its antimuscarinic mechanism of action. This mechanism antagonizes acetylcholine action at muscarinic receptors, both centrally and peripherally” – as above, plus the degree of description here suggests that the mechanisms are not elusive?

Section 2.1.3.

·       “Dextroamphetamine is a sympathomimetic drug that mimics the effect of endogenous agonists of the sympathetic nervous system” – I don’t know what this means

Section 2.1.4.

·       “Glucocorticoids such as dexamethasone (serotonin receptor antagonist)” (also Table 1) – glucocorticoids are not serotonin receptor [receptor not stated] antagonists

Table 1. Does not contain the information discussed in the text. Maybe the text could be changed (to introduce critical comment?) and the Table expanded to include the details?

Nutrition – no mechanisms discussed. For example, activation of Methol/Icilin receptors? Fat-induced ileal break? Please try to suggest mechanisms

Many cited studies have been conducted using rats and mice. These animals cannot vomit and their capacity to perceive nausea (measured only by human verbalisation) is uncertain. Consequently, mechanisms of action may not translate to humans. The authors need to point out this big limitation.

Music – were there any relaxation controls? Please be critical.

Discussion. “It has been shown that the efficiency of medications such as Cyclizine, Dramamine, Meclizine, Promethazine, Cinnarizine, and Dextroamphetamine, is because of their anti-histamine activity”. So far as I’m aware, dextroamphetamine has no anti-histamine activity. Further, these drugs have additional activity (eg muscarinic receptor antagonism) – how would the authors separate these actions?

Additional small points

·       Fig 2. N&V pointing to small intestine – may be more appropriate to point at stomach?

·       Introduction, ppg 4. Better to say “existing medications” since in the future, this statement may not be true?

Author Response

(The authors gave the same response as above.)

Round 2

Reviewer 1 Report

After the author's revision, this article seems to have improved significantly.

Author Response

We appreciate the time and effort you dedicated to reviewing our manuscript. Your comments have been incredibly helpful in enhancing the quality of our work. Thank you very much.

Dr. Abdullatif Tay

Director

Food Safety and Global Process Authority

433 W Van Buren St.

Chicago, IL 60607

Tel: 847-421-9741 (Cell)

Reviewer 2 Report

Antihistamines

Diphenhydramine and cyclizine are both H1 receptor antagonists and both inhibit cholinergic function. So why do they appear to have different actions (your reference 52)?

Betahistine is said to have "multiple effects on histamine receptors". What effects? What receptors? Similarly cinnarizine is said to have antihistamine actions but its not said what these are. The same for meclizine (what receptors? what other actions?). Previously it was the word "SLIGHT" anticholinergic and sedative effects which worried me. What does "slight" mean (eg can you quantify?)?

Anticholinergics

M3/5 most likely involved (Golding & Stott 1997; Br J Clin Pharmacol 43, 644; Soto & Vega 2010 Curr Neuropharmacol 8, 26).

Plant based interventions

Ginger is said to alleviate nausea by impeding gastric dysrhythmia and a rise in vasopressin. However these are the consequences of all treatments which reduce nausea and not a mechanism of action. You have a better mechanism described in the Discussion (although other possibilities do exist).

I do not understand how protein meals reduce gastric dysrhythmia by increasing gastrin secretion (which regulates gastric acid section)

Are the sections on the effects of fat-based meals a repetition of what has previously been stated?

Discussion

1. Ppg 3. What is the "vomiting epicentre"?

2. Section 3.1.3. Statement not referenced. Reviewed by Sanger & Andrews 2018. Front. Pharmacol 9:913 (or cite references within)

Typos

1. Introduction, ppg 3. Beginning of new addition: VIMS does not need to be in parenthesis. Second sentence of same section:' in' not 'n'

2. Histamine H1 should be written with the 1 in lower case. Similarly, drugs are histamine receptor antagonists not histamine antagonists. Trade names (eg Dramamine) begin with a capital letter but chemical names (eg cyclizine) do not.

Author Response

(The authors gave the same response as above.)
